# When Outcomes Diverge: Age and Cardiovascular Risk as Determinants of Mortality and ICU Admission in COVID-19

**DOI:** 10.3390/jcm11144099

**Published:** 2022-07-15

**Authors:** Marco Ranucci, Gianfranco Parati, Umberto Di Dedda, Maurizio Bussotti, Eustachio Agricola, Lorenzo Menicanti, Sara Bombace, Fabiana De Martino, Stefano Giovinazzo, Antonella Zambon, Roberto Menè, Maria Teresa La Rovere

**Affiliations:** 1Department of Cardiovascular Anesthesia and ICU, IRCCS Policlinico San Donato, 20097 San Donato Milanese, Italy; umbertodidedda@gmail.com; 2Department of Cardiovascular Neural and Metabolic Sciences, Department of Medicine and Surgery, IRCCS Istituto Auxologico Italiano, University of Milano-Bicocca, 20126 Milan, Italy; gianfranco.parati@unimib.it; 3Department of Cardiac Rehabilitation, IRCCS Istituti Clinici Scientifici Maugeri, 20138 Milan, Italy; maurizio.bussotti@ics.maugeri.it; 4Cardiovascular Imaging Unit, Cardio-Thoracic-Vascular Department, Vita-Salute University, IRCCS San Raffaele Hospital, 20132 Milan, Italy; agricola.eustachio@hsr.it; 5Scientific Directorate, IRCCS Policlinico San Donato, 20097 San Donato Milanese, Italy; lorenzo.menicanti@grupposandonato.it; 6Department of Biomedical Sciences, Humanitas University, Pieve Emanuele, 20090 Milan, Italy; sara.bombace@humanitas.it; 7IRCCS Humanitas Research Hospital, 20089 Rozzano, Italy; 8Heart Failure Unit, Centro Cardiologico Monzino Istituto di Ricovero e Cura a Carattere Scientifico, 20138 Milan, Italy; fabiana.demartino@cardiologicomonzino.it; 9Cardiovascular Disease Unit, Cardiac, Thoracic and Vascular Department, IRCCS Ospedale Policlinico San Martino, 16132 Genova, Italy; stefano.giovi88@gmail.com; 10Biostatistics Unit, IRCCS Istituto Auxologico Italiano, 20126 Milan, Italy; antonella.zambon@unimib.it; 11Department of Statistics and Quantitative Methods, Università di Milano-Bicocca, 20126 Milan, Italy; 12Department of Cardiology, IRCCS Istituto Auxologico Italiano, Università di Milano-Bicocca, 20126 Milan, Italy; r.mene@campus.unimib.it; 13Department of Cardiac Rehabilitation, IRCCS Istituti Clinici Scientifici Maugeri, 28843 Montescano, Italy; mariateresa.larovere@icsmaugeri.it

**Keywords:** COVID-19, heart failure, cardiovascular risk factors

## Abstract

Background: Hospital mortality and admission to the Intensive Care Unit (ICU) are markers of disease severity in COVID-19 patients. Cardiovascular co-morbidities are one of the main determinants of negative outcomes. In this study we investigated the impact of cardiovascular co-morbidities on mortality and admission to the ICU in first-wave COVID-19 patients. Methods: A multicenter, retrospective, cohort study. A total of 1077 patients were analyzed for mortality and ICU admission. Cardiovascular risk factors were explored as determinants of the outcomes after correction for other confounders. Results: In the multivariable model, after correction for age, only a history of heart failure remained independently associated (*p* = 0.0013) with mortality (hazard ratio 2.22, 95% confidence interval 1.37 to 3.62). Age showed a mortality risk increase of 8% per year (hazard ratio 1.08, 95% confidence interval 1.05 to 1.10, *p* = 0.001). The transition from ward to the ICU had, as a single determinant, the age, but in a reversed fashion (hazard ratio 0.96, 95% confidence interval 0.94 to 0.98, *p* = 0.0002). Conclusions: Once adjusted for the main determinant of mortality (age) heart failure only remained independently associated with mortality. Admission to the ICU was less likely for elderly patients. This may reflect the catastrophic impact of the first wave of COVID-19 pandemic in terms of ICU bed availability in Lombardy, leading to a selection process for ICU admission.

## 1. Introduction

Since the early reports on the COVID-19 pandemic, in-hospital mortality has been linked to the presence of a number of comorbidities. Predictors of mortality include respiratory pathologies (chronic obstructive pulmonary disease, asthma, smoking habits) metabolic diseases (diabetes, dyslipidemia, obesity), organ dysfunction (chronic renal failure, liver dysfunction), malignancies, and cardiovascular pathologies [1,2,3]. Among these, the most common are hypertension, coronary artery disease, valvular disease, heart failure, peripheral vascular disease, and cerebrovascular disease.

Even considering the role of comorbidities, the main predictor of mortality in COVID-19 patients is age. One of the first large studies reporting the outcome of COVID-19 patients admitted to the Intensive Care Unit (ICU) reported a 3-month survival rate of 35% in patients aged ≥64 years vs. 70% in those <64 years, with a relative increase in mortality of 86% per decade of age [4,5]. Cardiovascular risk factors (hypertension, hypercholesterolemia, heart failure, cardiomyopathy, or other heart disease) were all associated with hospital mortality. However, after correction for age, only hypercholesterolemia maintained a modest association with mortality [4].

Therefore, there is a gap of information with respect to cardiovascular risk factors as determinants of morbidity and mortality in COVID-19. The main uncertainty pertains to their role as independent determinants, or simply covariates expressing the usual comorbidities associated with age. Additionally, even their role in determining the severity of the disease, in terms of admission to the ICU, remains not investigated in large patient populations.

The present retrospective cohort study was designed to investigate if age and cardiovascular risk factor(s) are independently associated with mortality or admission to the ICU in COVID-19 patients.

## 2. Materials and Methods

### 2.1. Study Cohort

We considered patients hospitalized from March to June 2020 with COVID-19, recorded in the national multicenter registry of the “Cardiovascular risk and ancillary effects of cardiological drug therapy during CoV-19 infection” (“CARDICoVRISK” Study). This project involved 13 scientific institutes for research, hospitalization, and healthcare operating in Italy and was funded by the Italian Health Ministry. The study is registered at ClinicalTrials.gov (NCT04371289). This specific work was supported by a grant from Italian Ministry of Health (Ricerca Corrente Reti-Rete Cardiologica-2020 e 2021-RCR-2020-23670065 and RCR-2021-23671212). The Ethics Committee of the Centro Auxologico Italiano approved this study 2020_03_26_02 (approval date 30 March 2020).

The registry was built to collect information on patients diagnosed with COVID-19 (positive test for COVID-19 and positive chest x-rays and/or computer tomography scan for interstitial pneumonia compatible with infection) during their inpatient stay. Data collected included: (i) clinical, anthropometric, and medical history data (such as cardiovascular comorbidities); (ii) drug therapy before the onset of infection; (iii) clinical course of infection and outcome. Only patients without missing data in the main demographic and clinical variables were included. Each patient was followed until death or discharge. The original database included 2902 patients, but only 1073 patients had complete data for the analysis. Six institutions out of the 13 could not provide any patient with complete data, and, therefore, did not participate to this study. The main reason for the large number of patients with missing data is that during the first wave of the COVID-19 pandemic the clinical burden was overwhelming, and many institutions found it difficult to collect complete data.

### 2.2. Statistical Analysis

Continuous data were shown as mean and standard deviation (or median and interquartile range in the case of non-normally distributed data) and categorical data as absolute and relative frequencies. Comparisons between groups (alive and dead) were performed by means of t-test for independent sample (or Wilcoxon test in case of non-normally distributed data) for continuous covariates and chi-square test (or Fisher test) for categorical ones.

The univariate Cox proportional hazard regression model was fitted to estimate the association among each covariate, selected a priori by expert clinicians, and in-hospital death (hazard ratio—HR), its 95% confidence interval (CI) and *p*-value. To consider that proportional hazards assumption might not be realistic for all data, we performed stratified analysis by centers. The time-dependent nature of admission in intensive care during follow-up was considered [6]. A multivariate Cox proportional-hazard regression model was fitted to estimate the adjusted association estimates. Finally, we implemented a multi-state model considering the transition between three states: (i) admission in hospital, (ii) admission to intensive care, and (iii) in-hospital death (absorbent state) (Figure 1). This model allows us to estimate the separate effect of covariates on each transition. Following the approach of Putter et al. [7], we built a dataset in which each patient was repeated as many times as potential transitions. The standard error of the Cox model was obtained by robust sandwich estimators for the covariance matrix. In this model we included age and gender, and those covariates resulted statistically significant at the multivariate Cox regression model performed to identify the determinants of in-hospital death.

The SAS software was used for the analyses (SAS, Version 9.4; SAS Institute, Cary, NC, USA). For all hypotheses, the tested two-tailed *p*-values less than 0.05 were considered to be significant.

## 3. Results

### Description of the Cohort

A cohort of 1073 patients was analyzed in this study. Patient’s data were collected in 7 institutions. Six institutions out of seven collected patients from both the ICU and the ward, while one institution (Maugeri) only collected data from ward patients. Overall, in-hospital death accounted for 147 (13.7%) patients. Table 1 describes the demographics and cardiovascular risk factors of the patient population, with univariate association with mortality. With the only exception of gender, body mass index (BMI), history of deep venous thrombosis (DVT), diabetes, and valvulopathy, all other conditions were significantly associated with in-hospital mortality. ICU patients had a 33% mortality proportion; length of hospital stay was longer for the survivors.

Univariate and multivariable Cox regression models were applied to data reported in Table 1, producing HR and 95% CI (Table 2). In the univariate model, factors associated with in-hospital mortality were age (incremental relative risk 7% per year), previous myocardial infarction (HR 2.13), heart failure (HR 2.94) PVD (HR 2.80), and admission to the ICU (HR 3.08). In the multivariable model, the independent risk factors for mortality remained age (incremental relative risk 8% per year), previous myocardial infarction (HR 1.97), heart failure (HR 2.22), and admission to the ICU (HR 6.21).

The multistate model (Table 3) identified different roles of different factors depending on each transition. The transition from hospitalization to in-hospital mortality confirmed an incremental relative risk of 8% per year of age, a HR of 1.69 for myocardial infarction, and of 2.41 for heart failure. Transition from ICU to in-hospital mortality had only one independent factor (age, incremental relative risk 13% per year). Finally, the transition from hospitalization to ICU had only one independent factor (age). However, the impact of age with respect to this transition was opposite to the impact on in-hospital mortality. Whereas age was associated with an incremental relative risk of mortality of 8% (hospitalization to mortality) and 13% (ICU to mortality), in the transition from hospitalization to ICU admission there was a decreasing risk of 4% per year of age.

Figure 2 reports the different impact of age on the two outcomes considered (in-hospital mortality and ICU admission), with age acting as a risk factor for in-hospital mortality, but also as a protective factor for ICU admission. The reference line is the median age (71 years).

## 4. Discussion

The main results of our study are (i) age is the main determinant of in-hospital mortality in COVID-19 patients: once corrected for age, the only cardiovascular risk factors independently associated with in-hospital mortality are previous MI and heart failure; and (ii) ICU admission was determined by age, in a negative relationship, where every year of age above 71 years decreased the likelihood of ICU admission by 4%.

A considerable number of studies investigated the risk factors for mortality in COVID-19 in the years 2020–2021. A comprehensive article evaluating 20 systematic reviews and meta-analyses [2] identified a number of clinical risk factors for mortality. Cardiovascular risk factors included hypertension [8,9,10,11,12,13,14,15,16,17,18], cardiovascular diseases [8,11,14,16,17], coronary artery disease [9,12], heart failure [12], and cardiac arrythmias [16]. However, the great majority of these studies did not apply an age-corrected association of these risk factors with mortality.

Hypertension is almost invariably quoted within the cardiovascular risk factors. However, it is unclear whether or not this condition was controlled by specific therapies, if it was an anamnestic finding, or if it was diagnosed at the time of hospital admission. It is, however, well-known that arterial hypertension is an age-dependent disease, and our study confirms that, once corrected for age, the HR of hypertension is negligible (0.97) and even negatively associated with in-hospital mortality. There is an important study from Bhatia et al. [1], where the authors specifically addressed the role of hypertension as determinant of mortality in COVID-19 patients. Similar to our study, they found that hypertension had a prevalence of 45.6% in survivors and 71.6% in non-survivors (*p* < 0.001). However, in a multivariable model where age was included, hypertension yielded an odds ratio of 0.924 for mortality. Based on this and our results, we are inclined to consider arterial hypertension as a simple epiphenomenon of age, in the setting of COVID-19 associated morbidity and mortality.

The definition of “cardiovascular disease” contained in other previous studies is difficult to interpret and probably inclusive of many different conditions. Coronary artery disease was not an independent risk factor in a previous multivariable model [1]. Conversely, heart failure resists as independent risk factor for mortality in the same analysis, with an odds ratio of 2.712 (95% CI 1.127–6.526) absolutely in line with our findings (HR 2.22, 95% CI 1.37–3.62). A multivariable predictive model for mortality in COVID-19 patients included, as the only cardiovascular risk factor, congestive heart failure [19]. Of interest, in our series previous myocardial infarction maintained its independent role as a mortality predictor, albeit with a lower level of risk. Interaction between these two heart conditions could not be excluded. 

A large Italian experience from the most hit city during the COVID-19 pandemic (Bergamo) offers an important contribution to the analysis of mortality risk factors [20]. The authors analyzed 508 COVID-19 patients. They found an impressive number of co-morbidities significantly associated with mortality (about 34% of the cases). However, once pooled together in a multivariable analysis, only age and the severity of the disease (defined on the basis of ventilatory support and blood gas exchange) remained independent predictors of mortality.

As a conclusive remark to the role of cardiovascular factors as determinants of in-hospital mortality, we think that it is highly likely that only major cardiac co-morbidities and namely heart failure may be independent risk factors, the others being simply an epiphenomenon of advanced age.

The analysis of the factors leading to ICU admission brings to totally different considerations. Being a marker of severity of the disease, it is reasonable to think that the same factors leading to mortality could be associated with ICU admission. Conversely, the only independent factor associated with ICU admission is age, with a reversed behavior than in the previous analysis. In fact, the older the patient, the lower the likelihood of being admitted to the ICU. This finding deserves considerations that are more ethical and logistics than clinical.

Ninety-nine percent of our patient population was admitted in Lombardy Hospitals. Lombardy was the first region in Western Countries to be hit by the COVID-19 pandemic. In the early months of the pandemic, Lombardy Hospitals ICUs have been overwhelmed by the flow of COVID-19 patients with severe patterns of pneumonia requiring mechanical ventilation. At the peak of the pandemic, the ICU availability of beds was exceeded by 50% at least, with about 1500 patients requiring mechanical ventilation. At that stage, patients requiring minor forms of ventilatory assistance (namely non-invasive ventilation) were usually followed in the wards rather than in ICU. Under these stressful circumstances, the Italian Society of Anesthesia, Analgesia, Resuscitation and Intensive Care (SIAARTI) released a document with ethics recommendations for the allocation of intensive care treatments in exceptional, resource-limited circumstances [21]. Although age was not the only factor limiting the access to the ICU, the Ethics Committee of SIAARTI acknowledged that “An age limit for the admission to the ICU may ultimately need to be set. The underlying principle would be to save limited resources which may become extremely scarce for those who have a much greater probability of survival and life expectancy, in order to maximize the benefits for the largest number of people. In the worst-case scenario of complete saturation of ICU resources, keeping a “first come, first served” criterion would ultimately result in withholding ICU care by limiting ICU admission for any subsequently presenting patient” [21].

In fact, the “worst-case” scenario was rapidly reached with about 500 additional ICU beds gleaned from operating rooms, coronary units, cath labs and other locations. Many patients were intubated and mechanically ventilated in the emergency rooms, and remained there until a bed was available in the ICU; a centralized management of available ICU beds in Lombardy was settled in place on a 24 h 7/7 basis, and daily web-meeting among the Heads of ICU Departments were organized.

Under these conditions, it is highly likely that age being one of the main determinants of life-expectancy, the access to ICU beds was denied to elderly people in favor of much younger patients. 

The selective process of admission to ICU can be detected even in the already mentioned large Bergamo series [20]. Patients with severe respiratory pattern (PaO2/FiO2 < 200, N = 118) had an odds ratio of 3.5 for in-hospital mortality (*p* = 0.001) with a mortality rate of 57%. However, only 18 patients in this series received a tracheal intubation and mechanical ventilation, and quite surprisingly the mortality rate in this subgroup was 1.9%. Only 74 patients out of the 171 deceased received some form of advanced respiratory support (continuous positive airway pressure with helmet, non-invasive ventilation, or tracheal intubation), which means that 97 patients (20%) died without ICU admission.

Our data clearly reflected the sad decision making process of ICU admission during the first, dramatic COVID-19 pandemic wave. Age was clearly a negative predictor of ICU admission, and as many as 133 patients in our series died without passing through an ICU stay. 

There are many limitations in our study. The first and most important is the retrospective nature of the study and the large number of patients with missing data that led to the exclusion of many patients from the original database. As a consequence of the retrospective nature of the study, some variables were not collected and/or unretrievable: this applies to the pre-admission use of angiotensin receptor blockers and statins, and to the lack of information about the control of clinical conditions like hypertension, diabetes, and others. Another limitation is the relatively low sample size for patients admitted to the ICU. This suggests caution in interpreting the results of this stratum. Finally, given the ever-changing pattern of COVID-19 infection, our data collected during the first wave of COVID-19 pandemic are not necessarily replicable for the subsequent waves of the pandemic.

## 5. Conclusions

In conclusion, even if with divergent effects on the two outcomes considered, age remains the major player of the COVID-19 pneumonia outcome. Within this scenario, the only cardiovascular determinant of mortality was chronic heart failure.

## Figures and Tables

**Figure 1 jcm-11-04099-f001:**
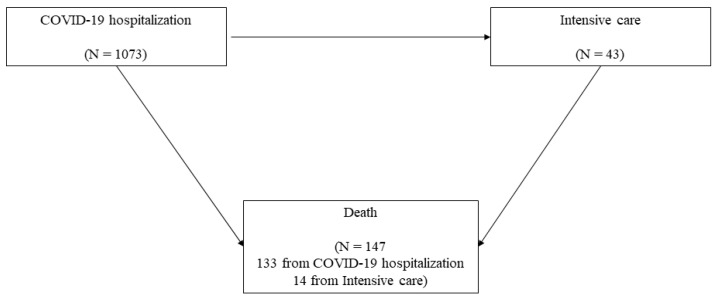
Multi-state model of transition between hospitalization, admission to the ICU, and mortality.

**Figure 2 jcm-11-04099-f002:**
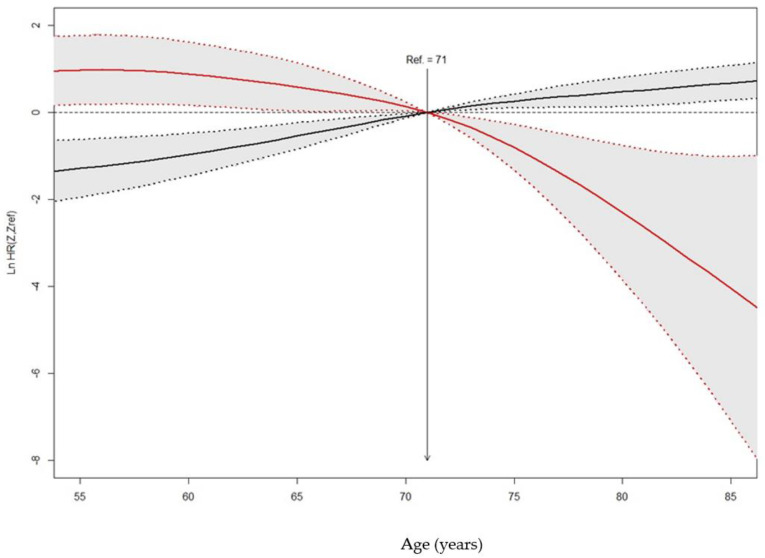
The likelihood of intensive care unit admission (red line) and mortality (black line) as a function of age. Dashed lines represent 95% confidence interval.

**Table 1 jcm-11-04099-t001:** Distribution of demographics and clinical characteristics of the whole cohort, alive, and death patients. Analysis of univariate association with mortality.

	Whole Cohort (N = 1073)	Alive (N = 926)	Death In-Hospital (N = 147)	*p*-Value
At Admission				
Center N (%)				
Humanitas	64 (6%)	44 (69%)	20 (31%)	
San Martino	2 (0%)	0 (0%)	2 (100%)	
Monzino	37 (3%)	26 (70%)	11 (30%)	
San Donato	237 (22%)	194 (82%)	43 (18%)	
Maugeri	539 (50%)	486 (90%)	53 (10%)	
Auxologico	73 (7%)	55 (75%)	18 (25%)	
San Raffaele	121 (11%)	121 (100%)	0 (0%)	
Age median [range IQ]	71 (61–80)	70 (59–79)	79 (72–85)	<0.0001 †
Gender N (%), Males	678 (63%)	576 (85%)	102 (15%)	0.0934 ‡
BMI Classes N (%)				0.1947 ‡
<25 Kg/m^2^	417 (39%)	350 (84%)	67 (16%)	
25–30 Kg/m^2^	403 (38%)	353 (88%)	50 (12%)	
>30 Kg/m^2^	253 (24%)	223 (88%)	30 (12%)	
History of PCI N (%)	33 (3%)	21 (64%)	12 (36%)	0.0008 *
History of CABG N (%)	27 (3%)	17 (63%)	10 (37%)	0.0018
History of Hypertension N (%)	636 (59%)	534 (84%)	102 (16%)	0.0072 ‡
History of MI N (%)	142 (13%)	109 (77%)	33 (23%)	0.0004 ‡
History of AF N (%)	63 (6%)	43 (68%)	20 (32%)	<0.0001 ‡
History of Heart failure N (%)	91 (8%)	57 (63%)	34 (37%)	<0.0001 ‡
History of Stroke N (%)	28 (3%)	20 (71%)	8 (29%)	0.0437 *
History of DVT N (%)	3 (0%)	1 (33%)	2 (67%)	0.0509 *
History of Valvulopathy N (%)	30 (3%)	22 (73%)	8 (27%)	0.0534 *
History of PVD N (%)	20 (2%)	11 (55%)	9 (45%)	0.0006 *
History of Cerebrovascular Disease N (%)	41 (4%)	31 (76%)	10 (24%)	0.0424 ‡
History of Diabetes (No Insulin) N (%)	210 (20%)	168 (80%)	42 (20%)	0.0031 ‡
History of Diabetes (Insulin) N (%)	166 (15%)	139 (84%)	27 (16%)	0.2958 ‡
Use of ACE Inhibitors N (%)	127 (12%)	100 (82%)	27 (18%)	0.0083 ‡
Use of Beta Blockers N (%)	155 (14%)	121 (78%)	34 (22%)	0.0013 ‡
During Hospitalization				
Intensive Care N (%)	43 (4%)	29 (67%)	14 (33%)	0.0002 ‡
Length Stay Median [range IQ]	22 (13–35)	24 (15–38)	11 (5–20)	<0.0001 †

† Wilcoxon test; ‡ Chi-square test; * Fisher test; ACE = Angiontensin-converting enzyme; AF = Atrial fibrillation; BB = Beta Blockers; BMI = Body mass index; CABG = Coronary artery bypass graft surgery; MI = Myocardial infarction; PCI = Percutaneous coronary intervention; PVD = Peripheral vascular disease.

**Table 2 jcm-11-04099-t002:** Unadjusted (univariate Cox regression model) and adjusted (multivariate Cox regression model), relative 95% confidence interval and *p*-value.

	Univariate Model		Multivariate Model	
	HR (95% CI)	*p*-Value	HR (95% CI)	*p*-Value
*At admission*				
Age	1.07 (1.05 to 1.09)	<0.0001	1.08 (1.05 to 1.10)	<0.0001
Gender *Male* vs. *Female*	1.10 (0.77 to 1.57)	0.6058	1.15 (0.79 to 1.69)	0.4679
BMI class				
25–30 vs. <25	0.70 (0.48 to 1.02)	0.699	0.73 (0.49 to 1.08)	0.1122
>30 vs. 25	0.61 (0.39 to 0.96)	0.609	0.66 (0.41 to 1.07)	0.0914
History of PCI *Yes* vs. *No*	1.40 (0.71 to 2.78)	0.3341	0.66 (0.29 to 1.50)	0.3218
History of CABG *Yes* vs. *No*	1.42 (0.68 to 2.96)	0.3491	0.97 (0.41 to 2.33)	0.9534
History of Hypertension *Yes* vs. *No*	1.17 (0.81 to 1.69)	0.3916	0.97 (0.65 to 1.46)	0.8840
History of MI *Yes* vs. *No*	2.13 (1.40 to 3.24)	0.0004	1.97 (1.23 to 3.16)	0.0046
History of AF *Yes* vs. *No*	1.31 (0.77 to 2.23)	0.3264	1.08 (0.61 to 1.89)	0.7985
History of Heart Failure *Yes* vs. *No*	2.94 (1.92 to 4.49)	<0.0001	2.22 (1.37 to 3.62)	0.0013
History of Stroke *Yes* vs. *No*	1.13 (0.54 to 2.38)	0.7241	1.49 (0.56 to 3.98)	0.4279
History of Valvulopathy *Yes* vs. *No*	1.12 (0.53 to 2.38)	0.7601	0.96 (0.42 to 2.19)	0.9177
History of PVD *Yes* vs. *No*	2.80 (1.25 to 6.27)	0.0124	2.25 (0.96 to 5.29)	0.0626
History of Cerebrovascular disease *Yes* vs. *No*	1.02 (0.53 to 1.99)	0.9483	0.74 (0.30 to 1.83)	0.5183
History of Diabetes (No Insulin) *Yes* vs. *No*	1.32 (0.90 to 1.93)	0.1562	1.03 (0.68 to 1.58)	0.8844
History of Diabetes (Insulin) *Yes* vs. *No*	1.18 (0.75 to 1.86)	0.4726	1.22 (0.73 to 2.04)	0.4494
Use of ACE inhibitors *Yes* vs. *No*	1.15 (0.75 to 1.78)	0.5250	0.94 (0.59 to 1.51)	0.8041
Use of BB *Yes* vs. *No*	1.03 (0.67 to 1.57)	0.9048	0.86 (0.52 to 1.44)	0.5705
During hospitalization				
Intensive Care *Yes* vs. *No*	3.08 (1.72 to 5.52)	0.0002	6.21 (3.21 to 12.03)	<0.0001

ACE = Angiontensin-converting enzyme; AF = Atrial fibrillation; BB = Beta Blockers; BMI = Body mass index; CABG = Coronary artery bypass graft surgery; CI: confidence interval; HR: hazard ratio; MI = Myocardial infarction; PCI = Percutaneous coronary intervention; PVD = Peripheral vascular disease.

**Table 3 jcm-11-04099-t003:** Adjusted association estimates (Hazard Ratio), relative 95% confidence intervals, and *p*-value obtained from a multistate model for each transition (“from Hospitalization to Intensive care”, “from Hospitalization to Death”, and “from Intensive care to Death”).

	From Hospitalization to Intensive Care		From Hospitalization to Death		From Intensive Care to Death	
	HR (95% CI)	*p*-Value	HR (95% CI)	*p*-Value	HR (95% CI)	*p*-Value
Age	0.96 (0.94 to 0.98)	0.0002	1.08 (1.06 to 1.10)	<0.0001	1.13 (1.03 to 1.23)	0.007
Gender *Male* vs. *Female*	1.69 (0.77 to 3.68)	0.191	1.26 (0.86 to 1.84)	0.244	1.00 (0.25 to 3.98)	0.998
MI *Yes* vs. *No*	0.29 (0.04 to 2.18)	0.230	1.69 (1.10 to 2.59)	0.017	NA	
Heart Failure *Yes* vs. *No*	1.15 (0.27 to 4.93)	0.848	2.41 (1.57 to 3.70)	<0.0001	0.58 (0.06 to 5.51)	0.639

CI = confidence interval; MI = Myocardial infarction; HR = hazard ratio; NA = Not Applicable.

## Data Availability

Data available on request and presently recorded in the IRCCS Cardiac Network RedCap Platform.

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
