# Peer review of "When Outcomes Diverge: Age and Cardiovascular Risk as Determinants of Mortality and ICU Admission in COVID-19"

_jcm, 2022, doi:10.3390/jcm11144099_

Round 1
Reviewer 1 Report
I read with interest the manuscript of Ranucci et al. Here are my few comments.
> in the description of the study cohort, the authors stated only patients without missing data have been included in the study. Can the authors precise how large is the whole cohort and how many patients have been excluded. Some lines later, in the Results section, it is writen that data has been collected in 7 institutions. Why the 6 others institutions (the project involved 13 institutions) didn't provide any data?
> Table 1, the authors analysed use of ACEi and use of betablockers between survivors and non-survivors. Why this drugs specificaly? Even if the protective role of ARB has now been denied, it would be as - or even more -interesting to analyse ARB than ACEi.
> Following the recent paper by Bouillon et al (https://www.ahajournals.org/doi/10.1161/JAHA.121.023357?url_ver=Z39.88-2003&rfr_id=ori:rid:crossref.org&rfr_dat=cr_pub%20%200pubmed), can the authors provide the differecne of statin use between the 2 groups ?
> Even if it could be obvious, the authors should emphasize the limitation of the retrospective design of the study.
Author Response
See the attcahed Word file

Reviewer 2 Report
In this retrospective cohort study, Ranucci et al. investigated if age, cardiovascular risk factors and diseases where independently associated with mortality or admission to the ICU in COVID-19 patients during the first wave of the pandemic.
The work is interesting, the methodology is sound, and the results are well presented. My only concern is that it is now a rather dated work: after the first wave, many others have come along, with several variants of Sars-CoV2 occurring and progressive shifts in the management of the pandemic. Although the clinical applicability of the study results in the contemporary scenario may therefore be affected, this is not critical to me. I believe the work adds valuable information to the existing knowledge and may particularly shed some light on a still controversial aspect of the pandemic, namely the role of cardiovascular risk factors and diseases as independent determinants of in-hospital death and admission to ICU. Anyway, the above issue should still be mentioned in the limitation section, in my opinion.
Other minor comments are listed below:
- I strongly advise the authors to revise the text to identify any typos and improper expressions (e.g., "bad outcomes," "nice study," etc.).
- In the introduction, the sentence " and the clinically relevant pertains their role as independent determinants or simply covariates expressing the usual comorbidities associated with age" is rather unclear and should be reworded.
- Since the control of cardiovascular risk factors at the time of hospitalization is not known (acceptable blood pressure control? acceptable glycemic control? etc.) I think this should be mentioned in the limitation section.
- Title of Table 1 mentions the results of the univariate analysis in terms of HR and 95%CI but these are not shown in the table (I guess it refers to Table 2).
Author Response
See the attached Word file
